# Direct translation of incoming retroviral genomes

Julia Köppke [1], Luise-Elektra Keller[1,3], Michelle Stuck[1,4], Nicolas D. Arnow[1], Norbert Bannert[1], Joerg Doellinger[2] & Oya Cingöz [1] ✉

Viruses that carry a positive-sense, single-stranded (+ssRNA) RNA translate their genomes soon after entering the host cell to produce viral proteins, with the exception of retroviruses. A distinguishing feature of retroviruses is reverse transcription, where the +ssRNA genome serves as a template to synthesize a double-stranded DNA copy that subsequently integrates into the host genome. As retroviral RNAs are produced by the host cell transcriptional machinery and are largely indistinguishable from cellular mRNAs, we investigated the potential of incoming retroviral genomes to directly express proteins. Here we show through multiple, complementary methods that retroviral genomes are translated after entry. Our findings challenge the notion that retroviruses require reverse transcription to produce viral proteins. Synthesis of retroviral proteins in the absence of productive infection has significant implications for basic retrovirology, immune responses and gene therapy applications.

All viruses, regardless of their nucleic acid type, composition or orientation, need to reach the mRNA stage for successful infection. Retroviruses carry two copies of a positive-sense, single-stranded RNA (+ssRNA) genome; however, they form a separate class from +ssRNA viruses in the Baltimore classification because their replication strategy involves reverse transcription of the ssRNA genome into a double-stranded (dsDNA) copy. Based on current knowledge most if not all +ssRNA viruses directly translate their RNA to synthesize viral proteins upon entry into host cells, with the exception of retroviruses, which undergo reverse transcription and degrade the original genomic RNA in the process. As retroviral genomes are produced by the host cell machinery, with a 5′ cap and a 3′ poly-A tail, we asked: Are incoming retroviral genomic RNAs also directly translated?

One small but notable difference between the full-length retroviral RNA packaged into the virions as the genome and the one that produces Gag and GagPol polyproteins is their transcription start sites (TSS). Full-length retroviral RNAs contain heterogenous TSS, which alters the RNA structure and impacts its dimerization and translation potential[1–3]. While 1G transcripts are primarily dimerized and

selectively packaged into virions, 2G and 3G transcripts exist mainly as monomers and are enriched in polysomes[1,2]. In addition, incompletely-spliced retroviral transcripts carry hypermethylated caps bound by NCBP3 instead of eIF4E, and translated in an mTOR-independent manner, although such hypermethylated caps were not detected in virion-packaged RNA[4]. As genomic RNA isolated from virions can be translated in in vitro systems to produce retroviral proteins[5–7], the virion-packaged genomic RNA does not appear to be inherently untranslatable, even if it differs from the newlyproduced, full-length RNA that is translated to produce viral structural proteins.

Using multiple complementary approaches, including post-translational regulation of protein stability coupled to sensitive reporter assays, immunoprecipitation, polysome fractionation and SILAC-based mass spectrometry, we demonstrate here that incoming retroviral RNA genomes are translated shortly after cellular entry. This is a general process that occurs under a variety of conditions; with different viral genome lengths, cellular entry pathways and cell types. Extensive controls including nuclease treatments, checking for DNA or protein transfer, using fusion-defective viruses, and omission of

[1]Robert Koch Institute, Department of Infectious Diseases, Unit of Sexually Transmitted Bacterial Pathogens and HIV (FG18), Berlin, Germany. [2]Robert Koch Institute, Centre for Biological Threats and Special Pathogens, Proteomics and Spectroscopy (ZBS6), Berlin, Germany. [3]Present address: Institute of Cardiovascular Regeneration, Goethe University Frankfurt, Frankfurt, Germany. [4]Present address: Department of Chemistry, Heidelberg University, Heidelberg, Germany. ✉e-mail: cingoezo@rki.de

various viral components (e.g., *env*, *gag*, packaging signal) confirm that the signal is truly due to direct de novo translation from incoming genomes. Capsid mutants that display altered stability and uncoating kinetics impact the translation of incoming RNAs by regulating the access of the encapsidated RNA to the translational machinery. The synthesis of retroviral proteins in the absence of reverse transcription has significant implications for basic retrovirology, immune responses during infection, and the use of retroviral vectors as RNA delivery vehicles.

## Results

### Post-translational control of protein stability minimizes virion-packaging and cellular delivery of reporter proteins

In the laboratory, retroviruses are typically produced by transfection of producer cells with plasmids encoding viral components. In case of viral genomes that carry a reporter gene, the reporter protein is also expressed in producer cells, which can get packaged into virions and delivered into recipient cells, yielding false positives[8–13]. To minimize such producer cell background, which could mask the signal from de novo translated incoming retroviral RNAs, we employed a post-translational protein control system (ProteoTuner), where a destabilizing domain (DD) derived from a cellular gene with a very short half-life (FKBP12) is fused to the gene of interest, targeting it for rapid proteasomal degradation[14]. The unstable protein can be stabilized in a dose-dependent and reversible manner by adding a cell-permeable small molecule ligand called Shield1 that binds to the DD, allowing post-translational regulation of protein levels. We reasoned that producing viruses in the absence of the ligand would minimize reporter protein packaging into virions, whereas performing infections in the presence of the ligand would allow us to detect reporter expression from the incoming retroviral RNA genomes by stabilizing the reporter. As only two genomic RNAs per retrovirus particle are delivered into the cell upon entry, we selected the sensitive reporter nano-luciferase (Nluc) to assay the translation of incoming retroviral RNA genomes, which has superior sensitivity compared to other luciferase proteins (reviewed in ref. 15).

Nluc was cloned with or without the DD either under a CMV promoter in a minimal lentiviral vector to produce pLVX-(DD)-Nluc, or with deletions in the CMV promoter, IRES element and Neo resistance gene to generate pLVX-(DD)-Nluc-ΔCIN (Fig. 1A). Transfection of these constructs into 293T cells resulted in 30–45-fold less luciferase activity in case of DD-harboring constructs, while Shield1 treatment rescued the expression (Fig. 1B). The stabilization of DD-Nluc constructs by Shield1 was dose-dependent, whereas Nluc constructs without the responsive domain remained unaffected (Fig. 1C). To quantify the amount of reporter protein packaging into virions, we produced virions by transfecting cells with the lentiviral transfer vectors shown in Fig. 1A along with a packaging construct with or without an *env* plasmid (VSV-G) and assayed the virus particles themselves for luciferase activity. Virions produced by DD-Nluc vectors consistently yielded 120–270-fold lower signal compared to Nluc vectors for both constructs, while the presence or absence of an envelope glycoprotein did not make a difference (Fig. 1D).

For HIV-1, the timing of the early events during infection are well-documented and cell type-dependent. Reverse transcription can take anywhere from 6 to 48 h, and integration follows about 5 h after the completion of reverse transcription[16,17]. As the constructs in Fig. 1A cannot produce the reporter protein from full-length (unspliced) viral RNA, they require reverse transcription, integration, transcription, splicing and translation to express luciferase. Accordingly, within the first 12 h of infection, no change in signal was observed, indicating that the luciferase signal represents Nluc protein that is transferred to and/or that remains associated with cells (Fig. 1E). The presence or absence of an Env protein or the RT inhibitor nevirapine (NVP) had no effect on

the signal observed, although infection with the destabilized reporter viruses consistently yielded less signal. Likewise, at 20 h post-infection, there was still no expression from the provirus, but expression did occur at 72 h post-infection (hpi), which was reduced to background levels in the absence of a functional Env or reverse transcription (Fig. 1F). These data show that post-translational regulation of protein stability can markedly reduce passive protein packaging into virions and their delivery into recipient cells, minimizing the background for assessing incoming retroviral RNA translation.

### Incoming retroviral genomes are used as an mRNA for protein expression early after entry

To investigate the direct translation potential of the full-length, incoming retroviral genomic RNA, we cloned Nluc with or without the DD downstream of the packaging signal (Ψ; psi) in place of where *gag* would normally be in a minimal lentivector. The start codon of *gag* was mutated and any additional elements that could affect translation efficiency (e.g., IRES, WPRE) were removed (Fig. 2A). In line with previous results, transfection of DD-constructs resulted in -180-fold decreased signal compared to Nluc, which was completely rescued by Shield1 (S1) addition (Fig. 2B). We produced virions and infected cells with serial dilutions of LV-DD-Nluc virus in the presence of NVP and S1, with or without the translation inhibitor cycloheximide (CHX) to distinguish virion-packaged protein delivery from de novo translation. Using viral doses as low as 0.0061 ng p24 per 100 K cells, we detected a CHX-sensitive signal (Fig. 2C). The signal to noise ratio (i.e., new protein synthesis vs. packaged protein delivery) was highest between 0.165 and 4.5 ng p24 per 100 K cells. Based on these results, we opted to use 1–4 ng p24 in subsequent experiments.

Infection of cells with LV-Nluc virus resulted in an increase in the luciferase signal over time despite the background of virion-packaged protein, which was diminished to background levels by CHX treatment (Fig. 2D). RT inhibition had no effect on the luciferase signal within the first 8 h, indicating that the translation occurs independently of reverse transcription. We then performed an infection with LV-DD-Nluc virus in the presence of NVP. The signal increased within the first 2 h, and under S1-stabilized conditions increased further up to 6 h, whereas in the absence of S1, it decreased after the initial hike due to the instability of the protein (Fig. 2E; red vs. blue lines). Translation inhibition by CHX drastically reduced the signal in both cases, although increased stabilization of the protein delivered passively by virions was also evident (Fig. 2E; green vs violet lines). Importantly, when DD-Nluc is stabilized, there is a clear increase in reporter expression over time, which is markedly reduced upon translation inhibition (Fig. 2E; red vs. violet lines). This difference is the result of newly synthesized reporter proteins from the incoming retroviral genomic RNA in the absence of reverse transcription. Similar results were observed regardless of whether the initial inoculum was kept on the cells or whether the virus was removed after 1 h, although the lack of continuous virus uptake in the latter case was evident (Fig. 2F, G).

### Expression from incoming retroviral genomes is due to de novo translation and occurs under different conditions

Viral supernatants typically contain plasmids carried over from transfected producer cells. Pre-treatment of our virus stocks with nucleases that degrade either DNA, RNA or both did not alter reporter expression from the incoming viral genome after transduction, despite the enzymes being functional, indicating that the observed signal is not due to nonspecific uptake of ambient DNA or RNA (Fig. 3A, B). The presence of a functional viral envelope on particles was necessary, as viruses pseudotyped with a VSV-G mutant defective in fusion activity (P127D) or those without an Env failed to yield a signal above background (Fig. 3C). Similar results were observed with Gag-less "virus" supernatants, when the packaging plasmid was omitted during virus

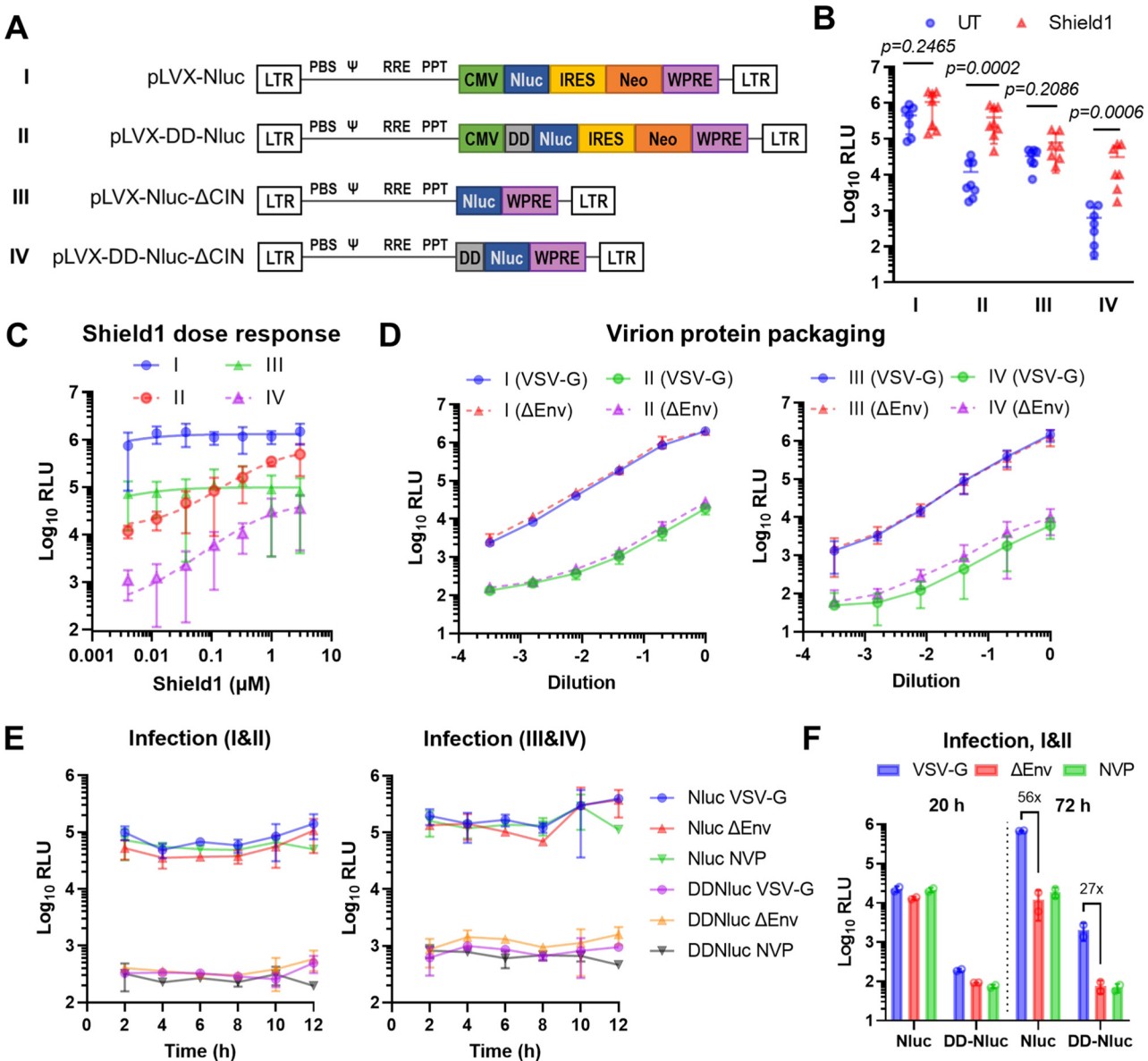

**Fig. 1 | Post-translational regulation of protein stability reduces protein packaging into lentiviral particles and their delivery into recipient cells.**
**A** Schematic representation of the constructs generated, not drawn to scale. Nanoluciferase (Nluc) was inserted with or without a destabilizing domain (DD) downstream of a CMV promoter in a minimal lentiviral vector to generate pLVX-(DD)-Nluc (constructs I and II). The CMV promoter, IRES element and Neomycin resistance gene were deleted to generate pLVX-(DD)-ΔCIN (constructs III and IV). **B** 293T cells were transfected with the indicated constructs in the presence or absence of Shield1. Nluc activity was measured after 24 h. **C** Transfection was performed as in (**B**) using different concentrations of Shield1. **D** Virus supernatants produced using the constructs in (**A**) with a packaging plasmid, with or without an Env (VSV-G) were assayed for virion-packaged Nluc activity. **E** Infection was performed with the viruses in (**D**) with or without NVP and assayed for Nluc activity at the indicated time points. **F** 293T cells were infected with pLVX-(DD)-Nluc carrying a VSV-G Env in the presence or absence of NVP, or with viruses without an Env. Nluc activity was measured at 20 and 72 hpi. IRES internal ribosomal entry site, WPRE woodchuck hepatitis virus posttranscriptional regulatory element, PBS primer binding site, psi (Ψ) packaging signal, RRE rev-response element, PPT polypurine tract, NVP nevirapine. Data are presented as means ± SD. Statistical analyses were performed by multiple Mann–Whitney *U* tests (unpaired, nonparametric, two-sided) using the false discovery rate (FDR) correction of Benjamini, Krieger and Yekutieli. Source data are provided as a Source Data file.

production (Fig. 3D). To ensure that our observations are not due to incomplete inhibition of RT, we confirmed these results with viruses containing a catalytic mutant RT (Fig. 3E and Supplementary Fig. 1A). To rule out endogenous reverse transcripts already present in virions or the inadvertent packaging and delivery of plasmid DNA fragments from producer cells as the source of the reporter signal, neither of which would be susceptible to nuclease digestion, we transduced cells in the presence of transcription or translation inhibitors (Actinomycin D [ActD] or CHX, respectively). Translation inhibition markedly reduced the signal as shown previously (Fig. 2C–G), whereas transcription inhibition did not, indicating that the observed signal is not due to DNA transfer, which would have required both of these processes (Fig. 3F). At the same concentrations both drugs inhibited expression from a transfected reporter plasmid, validating their functionality (Fig. 3G). Although inhibition of reverse transcription is not required for this process, we also confirmed these results during transduction in the presence of different RT inhibitors, namely NVP, efavirenz (EFV) and tenofovir (TAF) used at 10 μM (Fig. 3H), which resulted in similar expression levels as the untreated sample. When these RT inhibitors were used at higher concentrations (100 μM)

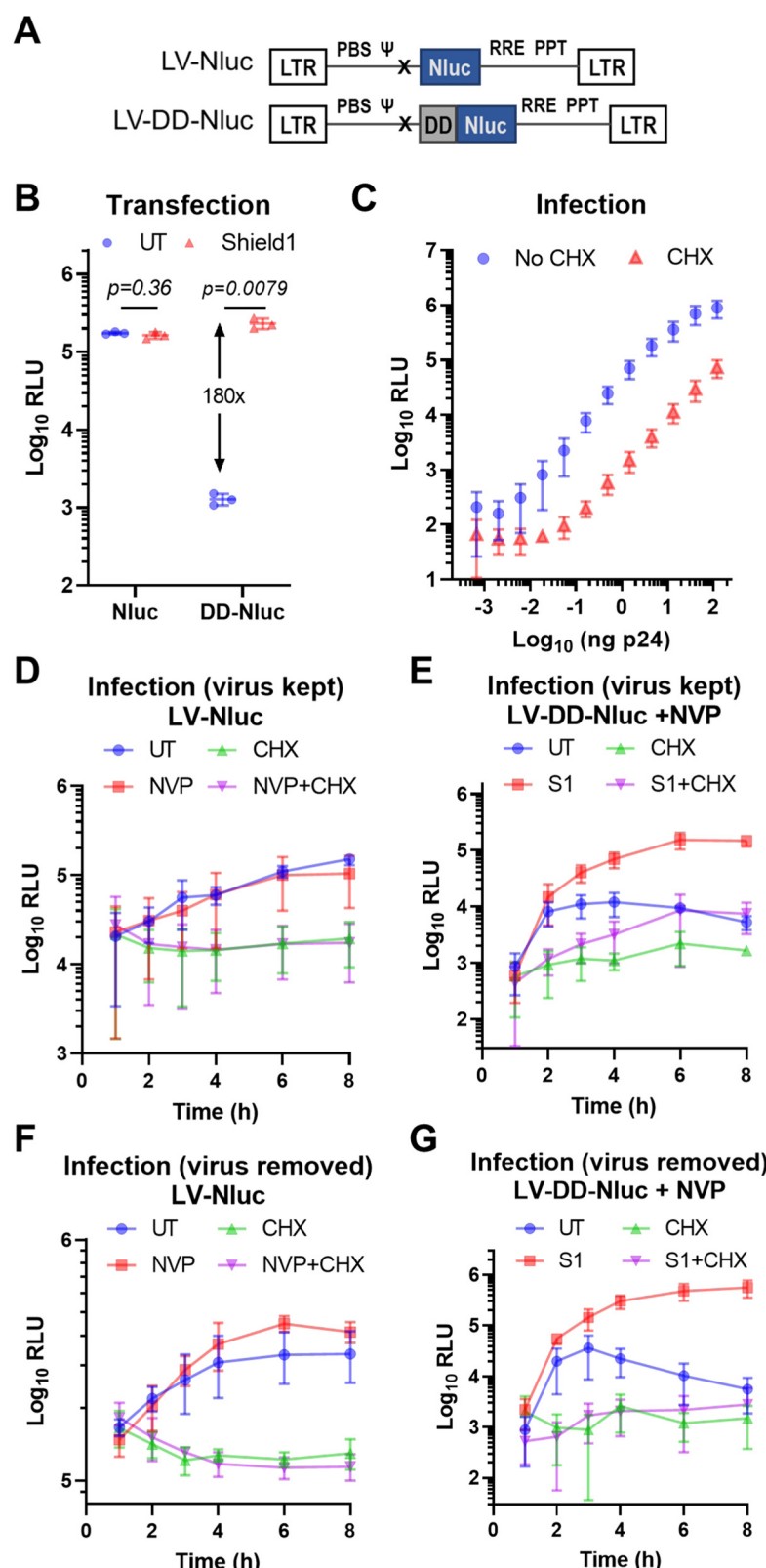

during infection, toxicity to cells was evident particularly in case of EFV, where cells also lost the ability to express a transfected plasmid (Supplementary Fig. 1B, C). Notably, the translation of incoming retroviral genomes was observed with viral envelopes that use different entry pathways (Fig. 3I) and in diverse cell lines and primary cells (Fig. 3J), indicating that this is a broad process not restricted to a specific cell type or condition.

**Immunoprecipitation, SILAC/MS and polysome fractionation confirm direct expression from incoming HIV-1 genomes**

The RNA genome of HIV-1 has an intricate secondary structure and several *cis*-acting RNA elements with diverse functions in the replication cycle, which are not all present in the context of a minimal vector[18]. To explore the contribution of such RNA elements to the process of incoming retroviral genomic translation, we cloned DD-

**Fig. 2 | Incoming minimal lentiviral RNA genomes are translated following entry into host cells. A** Schematic representation of the lentiviral constructs used; not drawn to scale. Nluc was cloned into a minimal lentiviral vector downstream of the packaging signal (Ψ), with or without the destabilizing domain (DD), and all other heterologous elements (CMV promoter, IRES, Neo resistance gene, WPRE) were removed. **B** Cells were transfected with the constructs in (**A**) in the presence or absence of Shield1 (S1). Nluc reporter activity was measured 1 day later. **C** Cells were infected with serial dilutions of LV-DD-Nluc in the presence of S1 and NVP, with or without cycloheximide (CHX), followed by Nluc measurement. **D, F** Cells were infected with VSV-G pseudotyped LV-Nluc virus in the presence or absence of NVP and/or CHX, where the virus was either kept on the cells (**D**) or washed away (**F**). Luciferase activity was measured at the indicated time points. **E, G** Infection was performed as in (**D**) and (**F**), but with LV-DD-Nluc and in the presence of NVP, with or without S1 and/or CHX. Data are presented as means ± SD. Statistical analyses were performed using two-sided, unpaired *t*-test with Welch's correction. Source data are provided as a Source Data file.

Nluc downstream of the packaging signal and a mutated *gag* start codon in a near-full-length HIV-1 construct based on NL4.3 that encodes firefly luciferase in place of *nef* (Fig. 4A). Infection with NL43-DD-Nluc, produced by co-transfecting cells with a packaging plasmid and a VSV-G *env*, yielded similar results to infection with minimal lentiviral vectors in the presence of NVP (Fig. 4B vs. Fig. 2E, G). Under S1-stabilized conditions the signal increased over time, peaking at 4–6 hpi, whereas translation inhibition reduced the reporter activity to baseline levels, confirming de novo protein synthesis (Fig. 4B; red vs. violet lines). Based on the relatively short incubation times, we did not remove the virus inoculum, which explains why the stabilized CHX condition shows a slight increase over time due to continuous re-entry (violet line).

To evaluate the number of RNA copies delivered into cells, we performed transductions with the parental NL43-Firefly virus in the presence or absence of NVP and quantified post-entry events at 24 and 48 h. As infection with 4 ng p24 per 100 K cells did not yield a detectable signal for RT products above background, we performed infections using 10 times more virus than normally used, which corresponds to a multiplicity of infection (MOI) of 1 based on the transducing units validated by p24 staining. Under these conditions, we detected early RT products between 4.1 and 8.2 copies, late RT products between 2.4 and 4.1 copies, and 2-LTR circles between 0.02 and 0.03 copies per haploid genome (Supplementary Fig. 2A).

As an alternative to reporter assays, we checked Gag protein production in the absence of reverse transcription. Cells were challenged with VSV-G-pseudotyped NL43-Firefly virus (MOI = 1) in the presence or absence of NVP and CHX. Gag production was assayed by immunoprecipitation (IP) with an anti-Gag polyclonal antibody and western blot 1 day after infection to maximize the accumulation of newly synthesized viral proteins. In the absence of RT inhibition, Pr55-Gag was produced at this time point, as expected. Although NVP treatment decreased this signal, viral protein production was still evident and did not originate from incoming CA protein, as CHX treatment abolished this signal (Fig. 4C).

To provide further evidence that authentic viral proteins are produced from the packaged viral genome, we performed metabolic labeling via stable isotope labeling with amino acids in cell culture followed by mass spectrometry (SILAC-MS). Briefly, cells grown in light medium were transduced with VSV-G-pseudotyped NL43-Firefly virus also produced in light medium (MOI = 0.5). At the time of transduction, cells were switched to heavy medium such that all newly produced proteins would be heavy, whereas all pre-existing proteins light. After 18 h, lysates were subjected to immunoprecipitation with an anti-p24 antibody followed by mass spectrometry. Under conditions where reverse transcription could take place, viral Gag production was evident, as cells had enough time to undergo the regular replication steps (Fig. 4D). Importantly, although the ratio of heavy to light peptides that map to Gag were lower in case of RT inhibition, such peptides were still detected, validating that incoming HIV-1 genomes are translated to produce viral proteins. Heavy peptides corresponding to Gag were only detected under conditions where translation could take place, confirming that these results truly are the consequence of de novo translation (Fig. 4D).

To validate the association of incoming HIV-1 RNA with polysomes - an indicator of active translation - we also performed polysome fractionation on lysates from 293T cells after 4 h of infection with VSV-G-pseudotyped IIIB with a catalytic RT mutation (DD185/186AA; Supplementary Fig. 1A). Infected and uninfected cells yielded comparable polysome profiles with clearly separated peaks for ribosomal subunits, whereas EDTA treatment completely disrupted polysomes (Fig. 4E). RT-qPCR on fractions demonstrated the association of 10% and 72% of all HIV-1 RNA and the housekeeping gene *HPRT1* (hypoxanthine phosphoribosyltransferase-1 RNA) with polysomes, which was reduced to 1% and 2% in EDTA-treated controls, respectively, indicating specific association of these RNAs with polysomes (Fig. 4F). Taken together, these data provide multiple independent lines of evidence that demonstrate the production viral proteins in the absence of reverse transcription.

## Packaging signal and capsid stability mutations impact the course of incoming retroviral RNA translation

The packaging signal (Ψ; psi) found in retroviral genomes is critical for the selective packaging of the full-length viral RNA into budding virions (reviewed in ref. 19). Deletion of 38 nucleotides (750–787) downstream of the Gag start codon in LV-DD-Nluc (ΔΨ) drastically reduced the amount of packaged genomic viral RNA compared to WT (Fig. 5A), even though the expression from the ΔΨ construct by transfection was comparable to WT with a 2.5-fold increase (Fig. 5B). Infection of cells with the two viruses showed significantly reduced reporter signal in ΔΨ compared to WT, indicating that the packaging of the viral genome into particles is essential for incoming viral RNA translation, as expected (Fig. 5C).

Since the viral RNA genome is protected by the capsid, we reasoned that mutations affecting capsid stability may alter incoming translation kinetics. To this end, we used two well-characterized capsid mutants; the hypostable P38A mutant that loses its integrity early after entry, and the hyperstable E45A mutant which does not easily dissociate. It was previously described that different titration methods show considerable variation in predicting the transducing units of lentiviruses[20]. We therefore normalized the viruses to each other by different methods: p24-CA amount by ELISA, viral genomic RNA by RT-qPCR, virion-packaged Nluc by luciferase assay, or simply by using equal volumes from viruses produced at the same time. Despite slight variations based on the normalization method used (Fig. 5D, E and Supplementary Fig. 3A, B), collectively, infection with the P38A mutant increased translation from incoming genomes compared to WT, whereas E45A mutant decreased it (Fig. 5F). These data are consistent with unstable capsids allowing increased access to the translational machinery, in contrast to hyperstable capsids shielding the viral RNA, linking capsid stability and uncoating kinetics to the translation potential of incoming retroviral genomes.

We show here that the RNA genome packaged into retroviral particles can serve as an mRNA for viral protein production independently of reverse transcription. This process occurs in the context of both minimal and near-full-length genomes, in the presence of RT inhibitors and catalytic mutants, with Env proteins that use different entry pathways and in different cell types with different virus amounts, suggesting that this is a general process. Using thorough controls that account for nucleic acid contamination, passive DNA or protein delivery, transcription and translation inhibitors, fusion-deficient envelopes and omission of various viral components, we confirm

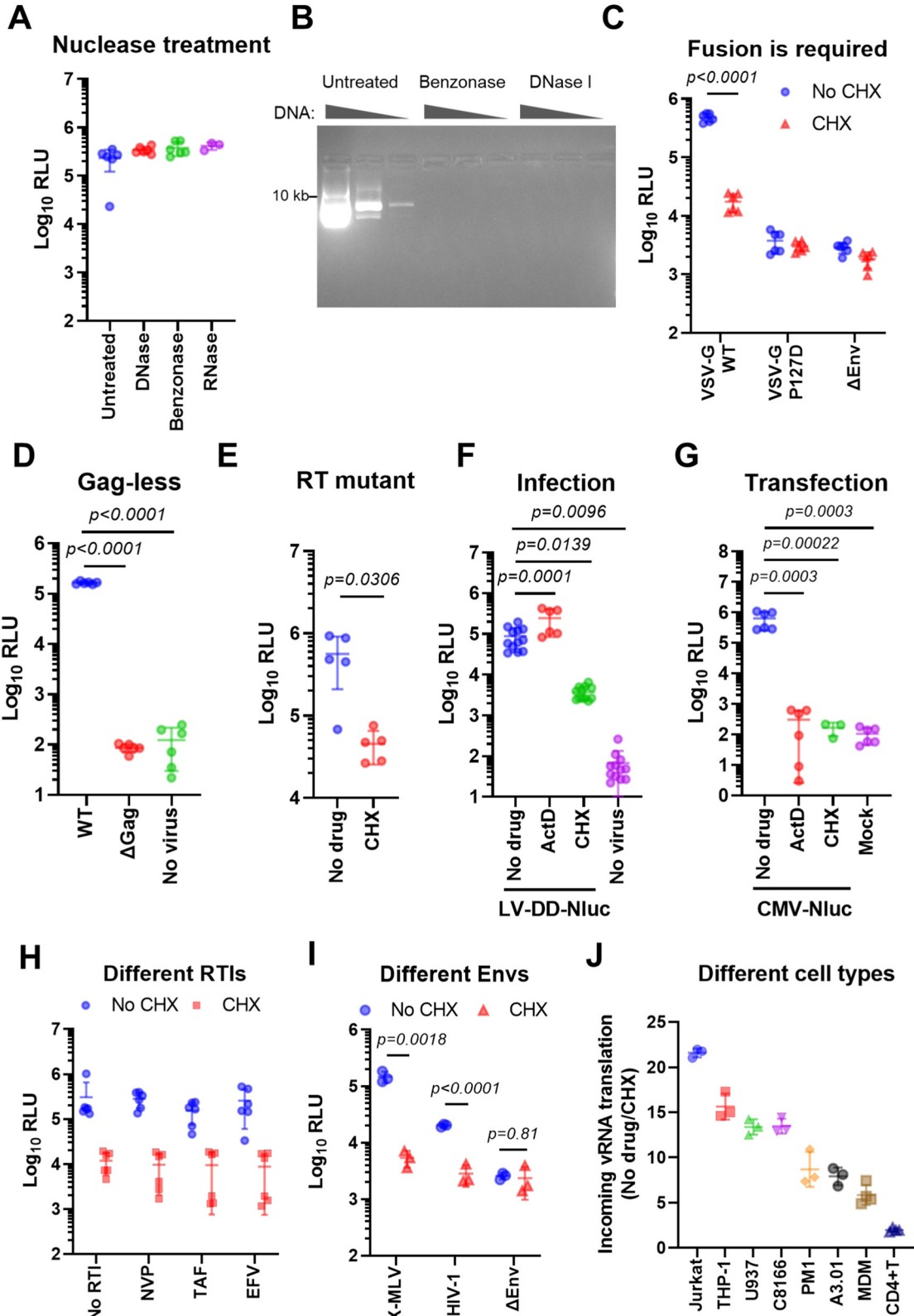

that the signal observed is due to de novo translation. Using capsid mutant viruses, we further demonstrate that the translation potential of the viral RNA is linked to capsid stability as the process of uncoating modulates the access of the incoming RNA to ribosomes. In summary, we provide multiple lines of evidence that the retroviral genome can serve as an mRNA early after cellular entry. Below we discuss potential implications of our findings for basic retrovirology, host immune responses and RNA delivery approaches.

## Discussion

Early studies of avian and murine retroviruses reported the detection of full-length viral genomes in polysome-containing pellets within 4

**Fig. 3 | Direct expression from incoming lentiviral RNA genomes is truly due to de novo translation and independent of reverse transcription.** Unless indicated otherwise, all infections were performed on 293T cells with LV-DD-Nluc virus in the presence of NVP and S1, with or without CHX, and assayed 4–6 h later. **A** Equal amounts of virus stocks were treated with nucleases or left untreated prior to transduction of cells. **B** Different amounts of plasmid DNA (0.1, 1 or 10 μg) were treated with nucleases or left untreated and visualized on an agarose gel. **C** Cells were transduced with viruses pseudotyped with WT VSV-G, a fusion-deficient mutant of VSV-G (P127D) or without an Env. **D** Transduction with "viruses" produced in the presence or absence of a lentiviral packaging vector (WT vs. ΔGag, respectively). **E** Transduction with reporter virus that carries a catalytic RT mutation (D110E) in the presence of Shield1. **F** Transduction in the presence of S1 and NVP, with or without ActD or CHX. **G** Transfection of cells with a plasmid encoding CMV-driven Nluc in the presence of ActD or CHX. **H** Transduction in the presence of different RT inhibitors at 10 μM. **I** TZM-bl cells were transduced with viruses pseudotyped with HIV-1 Env, X-MLV Env or no Env. **J** The indicated cell types were transduced with reporter viruses. Data are presented as means ± SD. Statistical analyses were performed by one-way ANOVA with Tukey's test (**A**, **D**, **F**, **G**) or by unpaired, two-sided *t*-test with Welch's correction (**C**, **E**, **I**). In case of multiple *t*-tests, the false discovery rate (FDR) correction of Benjamini, Krieger and Yekutieli were used (**C**, **I**). NVP Nevirapine, EFV Efavirenz, TDF Tenofovir Disoproxil Fumarate. Source data are provided as a Source Data file.

hpi[21–23]. A recent study also detected full-length HIV-1 transcripts associated with polysomes at 8 hpi; although it is unclear whether the expression at this time point is mediated by reverse-transcribed and integrated viral DNA that is then newly-transcribed, or from the incoming RNA genome itself, as the HIV-1 RT was functional in this case[24]. In support of our findings, a ribosome profiling study of HIV-1 infected cells detected ribosome-protected RNA fragments indicative of active translation in the *gag* coding region already within 1 h of infection[25]. Based on the current knowledge regarding infection kinetics, as 1 h is too short of a time period to complete reverse transcription and integration, the implication is that such expression is enabled by the direct translation of the RNA genome. Ribosome profiling studies in the absence of a functional RT at early time points following infection will be informative in identifying the specific regions of incoming retroviral genomes that are translated in different cell types.

In the field of gene therapy, recombinant retroviral vectors are very well-characterized and commonly used as tools for nucleic acid or protein delivery (reviewed in ref. 26). In a previous study using minimal gammaretroviral (MLV) vectors with primer binding site (PBS) mutations that cannot initiate reverse transcription, protein expression from a reporter gene was detected[27]. In other studies, modified lentiviral vectors containing 5′ IRES or 3′ WPRE insertions or major structural rearrangements of the genome (e.g., moving the U5-R regions further downstream towards the 3′ end) were employed in order to enable direct translation from the packaged RNA[28,29]. In all of these studies, however, protein or DNA transfer cannot be ruled out. In addition, to our knowledge, the production of actual retroviral proteins from a (near) full-length genome in the absence of reverse transcription has not been demonstrated.

A paradigm shift in retrovirology occurred with the finding that intact or near-intact HIV-1 capsids can be transported into the nucleus, where reverse transcription and uncoating is completed[30–33]. How to reconcile the direct translation of incoming retroviral genomes surrounded by capsid and inaccessible to the translational machinery with the detection of intact core structures in the nucleus? In a given virus population not all particles are infectious or replication-competent. In fact, for animal viruses the particle-to-PFU (or particle to infectious unit; P/IU) ratio can vary greatly; from 1 to 2 for Semliki Forest Virus to as high as 10^7 for HIV-1, according to some estimates (reviewed in ref. 34). Such a high ratio highlights the presence of a large number of particles that may not proceed successfully to next steps of the replication cycle. We believe that incoming retroviral RNA translation occurs in case of particles that are able to enter the host cell, carry a packaged genomic RNA and start uncoating before reaching the nucleus; an idea that has been employed as a basis for the EURT entry/uncoating assay[35]. As direct translation of the genome is likely a dead-end for viral replication, it is reasonable to assume that virions that end up producing infectious progeny are the few ones that successfully make it to the nucleus or the nuclear pore intact, not those that are translated. In the stochastic environment of a viral population, some genomes are translated, whereas some are reverse-transcribed, just as some capsids fall apart before reaching the nucleus while some of

them make it to the nucleus almost intact. As many of the assays used in this study are biochemical in nature, we cannot analyze the status of individual virus particles but rather the status of the population as a whole. Single-molecule live-imaging approaches to visualize incoming viral genomic RNAs and newly synthesized proteins will be valuable in quantifying the percentage of particles that are translated after cellular entry.

The production of viral proteins in the absence of reverse transcription could have major consequences. Individuals who encounter HIV-1 but who do not get productively-infected, for instance due to pre-exposure prophylaxis (PrEP) usage, may still be able to process and present viral peptides to generate cell-mediated and/or humoral immune responses. An initial abortive infection may result in the recruitment and activation of T-cells, increasing the eligible target cell population locally for productive infection. During SIV infection of rhesus macaques, Gag- and Pol-specific CTL responses were identified very early (within 2 h) after infection, whereas Env- or Nef-specific responses were not found until later, which was attributed to the ability of incoming viral proteins to be processed and presented[36,37]. Data presented here suggest that such responses may also occur due to de novo translation from the viral genome. In summary, our results challenge the notion that retroviruses require reverse transcription to produce viral proteins, warrant careful studies of immune responses during abortive infection and open up novel avenues for gene therapy and targeted vaccine approaches.

## Methods
### Cloning, constructs and virus production
As a basis for minimal lentiviral vectors the pLVX-IRES-Neo vector (Clontech) was used, which contains identical, full-length LTRs. DD-Nluc was synthesized as a gBlock (IDT) and cloned between XhoI and NotI sites of pLVX-IRES-Neo, either with or without the destabilizing domain to create CMV-(DD)Nluc. Similarly, (DD)Nluc was cloned between ClaI and MluI sites of pLVX-IRES-Neo to eliminate the CMV promoter, multiple cloning site, IRES element and Neo resistance gene to create (DD)Nluc-WPRE. LV-(DD)Nluc was created first by deleting the sequences between ClaI and KpnI sites (including CMV promoter, multiple cloning site, IRES element, Neo resistance gene and the WPRE sequence) in pLVX-IRES-Neo and ligating the vector back onto itself, then mutating the start codon of Gag, inserting the restriction sites BstBI and PacI after the packaging signal (Ψ) and finally cloning (DD)Nluc between these restriction sites. Packaging signal deletion (ΔΨ) was introduced into LV-DD-Nluc by overlap PCR resulting in a deletion between nucleotides 750–787 based on the pLVX-IRES-Neo vector sequence. NL43-Firefly is pNL4-3 e- r- FLuc (kindly provided by Ned Landau; NIH-ARP-3418). NL43-DD-Nluc was created by mutating the Gag start codon of NL43-Firefly and inserting the DD-Nluc sequence between PteI and SpeI sites. The identity of all constructs was confirmed by restriction digest and sequencing.

Reporter viruses were produced by transfecting viral plasmids together with a plasmid encoding VSV-G *env* (pCMV-VSV-G-myc, or the fusion-defective mutant P127D, kindly provided by Wes Sundquist; Addgene #80054 and #80055), and in case of the minimal lentiviral

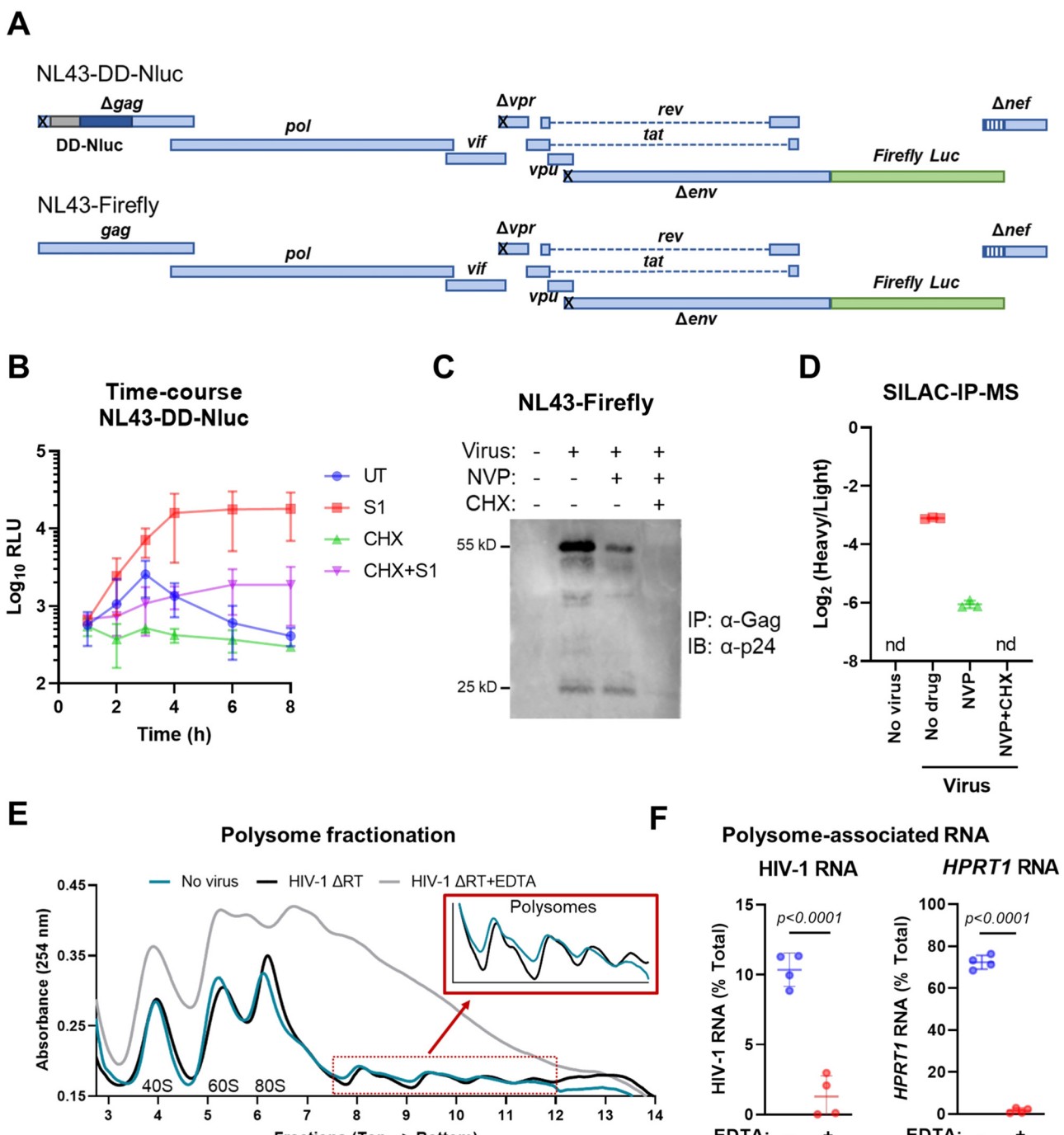

**Fig. 4 | Incoming near-full-length HIV-1 genomes are translated to produce viral proteins in the absence of reverse transcription. A** Schematic representation of the near-full-length HIV-1 constructs, where "x" denotes mutated codons. DD-Nluc was cloned into the NL43-Firefly construct downstream of the packaging signal to generate NL43-DD-Nluc. **B** 293T cells were infected with NL43-DD-Nluc with nevirapine (NVP) in the presence or absence of Shield1 (S1) and/or CHX. Nluc activity was measured at the indicated time points. **C** Cells were infected with VSV-G-pseudotyped NL43-Firefly with or without NVP or CHX. Lysates were collected 1 day after infection, immunoprecipitated with a polyclonal anti-HIV-1 Gag antibody and probed for p24. **D** Cells were labeled with heavy amino acids (SILAC) at the time of infection (MOI = 0.5) in the presence of the indicated drugs. 18 h later, Gag was immunoprecipitated from cell lysates and bound proteins were analyzed by mass spectrometry. Data are represented as the heavy to light ratio of peptides that map to Gag. **E** Representative polysome fractionation profiles from 293T cells infected (or not) with VSV-G pseudotyped IIIB ΔRT (with a catalytic RT mutation DD185/186AA) at 4 hpi and lysed in the presence or absence of EDTA. Lysates were run on a sucrose density gradient, then fractions were collected from top to bottom while simultaneously measuring UV absorbance. **F** RNA was isolated from each fraction and the levels of HIV-1 genomic RNA and *HPRT1* were measured by RT-qPCR. The amount of each RNA is given as a percentage of total RNA for that message. Data represent mean ± SD. Statistical significance was determined by unpaired, two-sided *t*-test with Welch's correction. nd not detected. Source data are provided as a Source Data file.

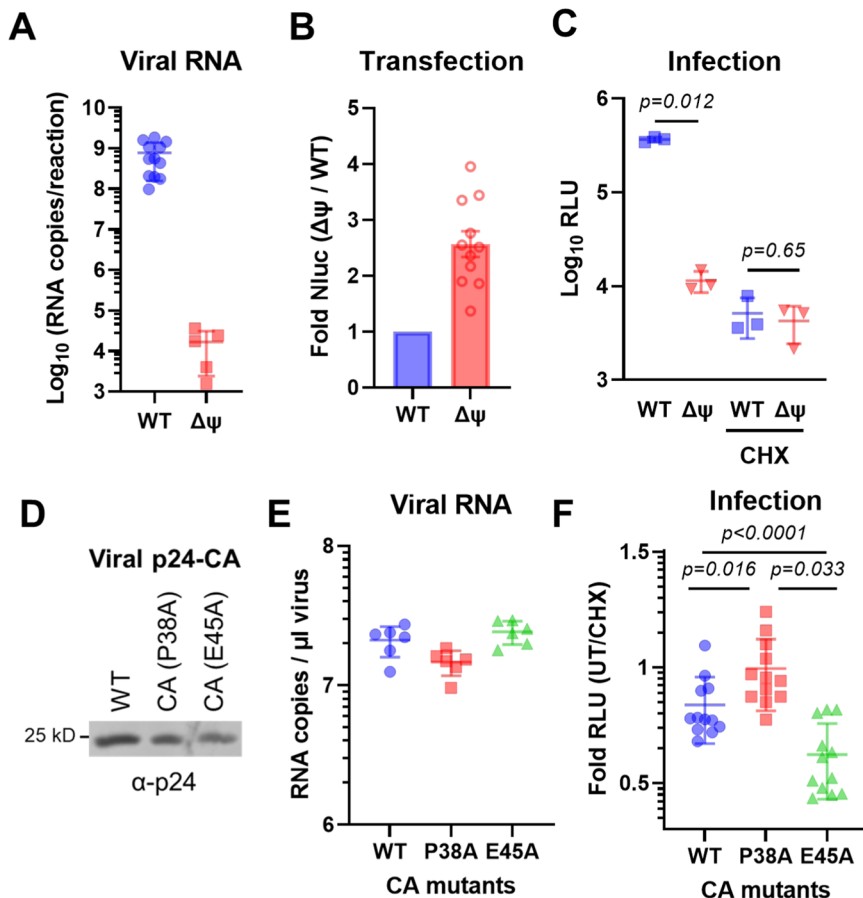

**Fig. 5 | Packaging signal and capsid stability affect the translation of incoming RNA genomes. A** Viral RNA content of LV-DD-Nluc viruses produced with or without a deletion between nucleotides 750–787 in the packaging signal (ΔΨ; delta psi) measured by RT-qPCR. **B** 293T cells were transfected with the LV-DD-Nluc WT or ΔΨ construct and subjected to luciferase assay a day later. **C** Cells were transduced with WT or ΔΨ viruses; expression from the incoming viral genome was analyzed by luciferase assay at 4–6 hpi. **D** Western blot of p24-CA in concentrated WT or the indicated capsid mutant virus stocks. **E** RT-qPCR for quantification of viral RNA. **F** Cells were transduced with reporter viruses carrying WT or mutated CA. All transductions were performed in 293T cells using LV-DD-Nluc viruses in the presence of both NVP and Shield1, with or without CHX. Data represent mean ± SD. Statistical significance was determined by unpaired, two-sided *t*-test with Welch's correction (**C**) or one-way ANOVA with Tukey's correction (**F**). Source data are provided as a Source Data file.

vectors or NL43-DD-Nluc, also with a packaging plasmid encoding HIV-1 gag, pol, tat and rev (psPAX2, kindly provided by Didier Trono; Addgene #12260) or with the packaging plasmid pCD/NL-BH*del-tavpu/RT- that lacks RT activity due to the D110E mutation in the catalytic site (kindly provided by Jakob Reiser; Addgene #136985). CA stability mutants P38A and E45A were kindly provided by Stephen Goff. The full-length IIIB ΔRT construct carrying the DD185/186AA mutations in the catalytic domain of RT was kindly provided by Michael Malim[38]. The HIV-1 *env* plasmid was kindly provided by Yiping Zhu (University of Rochester). X-MLV *env* was a codon-optimized, V5-tagged, synthetic Xenotropic MLV-Related Virus (XMRV) *env* in a pTH backbone, kindly provided by Alaa Ramadan[39].

Virus stocks were produced by transfection of 293T cells with polyethylenimine (PEI), followed by media change after 1 day and supernatant collection after 2 days. ΔEnv or ΔGag viruses were produced by omitting the respective plasmids during the transfection step. Virus-containing supernatants were filtered (0.45 μ), ultracentrifuged over a 20% sucrose cushion, aliquoted and frozen at −80 °C. Nuclease treatment of virus stocks was performed at room temperature by DNase (Ambion), RNase (Qiagen) or benzonase (Millipore).

### Cell culture, treatments, transductions

293T (F, CVCL_0063), Jurkat (M, CVCL_0065), and (CEM)A3.01 (F, CVCL_6244) were from ATCC. TZM-bl (F, CVCL_B478) and PM1 (M, CVCL_9472) were from the NIH HIV reagent program. THP-1 (M, CVCL_0006), U937 (M, CVCL_0007) and C8166 (M, CVCL_1099) were from internal laboratory stocks. Buffy coats of anonymous healthy donors were from the German Red Cross in compliance with all ethical regulations. 293T and TZM-bl cells were maintained in DMEM containing 9% FBS (Gibco) and 100 μg/ml Pen/Strep (Gibco). All suspension cell lines including THP-1, U937, C8166, Jurkat, PM1 and A3.01 were maintained in RPMI with 9% FBS, 100 μg/ml Pen/Strep, 100 μg/ml Normocin (Invivogen). PBMCs were isolated from the buffy coats using standard Ficoll separation. Monocytes were selected by adhering PBMCs in RPMI with 5% pooled AB human serum (Sigma), 1 mM HEPES (Gibco) and 24 μg/ml gentamicin (Sigma) for several hours, followed by extensive washing to remove unbound cells. Monocyte-derived macrophages were differentiated from primary monocytes by 50 ng/ml GM-CSF (R&D Systems) treatment for 6–10 days. CD4+ T-cells were isolated from PBMCs by negative selection using the MACS human CD4+ T-cell isolation kit (Miltenyi). Infections were performed by spinoculation at 1200 × g for 60–90 min at 25 °C. Unless indicated otherwise, the following concentrations were used for treatments: NVP: 10–25 μM (Merck), CHX: 10–100 μg/ml (EMD/Millipore), ActD: 2 μg/ml (Sigma), Shield1: 1.5–3 μM (Takara). With the exception of time-course experiments, Nluc activity was measured after 4–6 h post-infection using the Nano-Glo luciferase assay system (Promega).

**Table 1 | List of primers and probes used in the study**

| Target | F/R/P | Sequence | Reference |
|---|---|---|---|
| RT products | F | TGTGTGCCCGTCTGTTGTGT | 47 |
| | R | GAGTCCTGCGTCGAGAGATC | |
| | P | CAGTGGCGCCCGAACAGGGA | |
| 2-LTR circles | F | AACTAGGGAACCCACTGCTTAAG | |
| | R | TCCACAGATCAAGGATATCTTGTC | |
| | P | ACACTACTTGAAGCACTCAAGGCAAGCTTT | |
| HIV-1 genome | F | TCTCGACGCAGGACTCG | 48 |
| | R | TACTGACGCTCTCGCACC | |
| | P | CTCTCTCCTTCTAGCCTC | |
| HPRT1 mRNA | F | TCTTTGCTGACCTGCTGGATT | |
| | R | TTATGTCCCCTGTTGACTGGT | |
| | P | AGTGATAGATCCATTCCTATGACTGT | |

*F* forward, *R* reverse, *P* probe.

## Polysome fractionation

Polysome fractionation was performed as previously with some modifications[40]. Cells were split the day before such that they would reach a maximum of 60–80% confluency on the day of the assay. At 4 h post-infection with RT-deficient viruses or with WT viruses in the presence of NVP treatment, cells were treated with 50 μg/ml CHX for 10 min at 37 °C, washed once with ice-cold PBS+CHX (50 μg/ml), collected by scraping in PBS+CHX, pelleted, and lysed in 1X polysome lysis buffer containing 20 mM Tris-HCl pH 7.5, 150 mM KCl, 5 mM MgCl$_2$, 0.5 % NP40, 1 mM DTT, 50 μg/ml cycloheximide, and protease inhibitors (Roche) on ice for 10 min. Lysates were passed through a 21-gauge needle 12 times, incubated on ice for another 5 min and cleared by spinning at 4 °C, 15,900 × g for 10 min. Cleared lysates were then loaded on 15–45% linear sucrose gradients (in 20 mM Tris-HCl pH 7.5, 150 mM KCl, 5 mM MgCl$_2$, 50 μg/ml CHX) prepared using a BioComp gradient master in ultra-clear centrifuge tubes (Beckman Coulter) and centrifuged for 2:30 h at 160,000 × g in an SW41 rotor. Fractions (0.5 ml) were collected by a piston gradient fractionator (BioComp) with continuous UV absorbance recording at 254 nm. RNA from each fraction was isolated by phenol-chloroform extraction and quantified by RT-qPCR.

## qPCR, RT-qPCR

Viral genomic RNA was isolated from concentrated virus stocks with viral RNA isolation kit (Qiagen). RNA was treated with Turbo DNase and inactivation beads (Ambion), cDNA was synthesized using Superscript III (Invitrogen) or RevertAid RT (Thermo) with random hexamers (Roche). qPCR was performed with the SensiFAST No-ROX Probe Master Mix (Bioline) on a CFX96 qPCR machine (Bio-Rad) along with standards. Primer and probe sequences used in RT-qPCR are listed in Table 1.

## Immunoprecipitation, western blots, SILAC labeling

Immunoprecipitations were performed as described with slight modifications[41]. Protein G Dynabeads (Invitrogen) were washed and coated with anti-p55+p24+p17 antibody (Abcam) by rotating for 15 min at 4 °C. After washing off unbound antibody, cleared cell lysates were added to the coated beads and incubated at 4 °C with rotation for 1 h. Beads were then separated by a magnet, washed three times and bound proteins were released from the beads by boiling in the presence of a denaturing loading buffer. Released proteins were analyzed by SDS-PAGE and western blot based on previous protocols[42]. Briefly, cells were washed with PBS, scraped,

transferred to a tube and washed again with PBS. The pellet was lysed in NP40 lysis buffer (100 mM Tris, 30 mM NaCl, 0.5% NP40) containing benzonase for 15–30 min on ice. Lysates were cleared by centrifugation at 9400 × g for 5 min, supplemented with denaturing loading buffer (Invitrogen) and run on an SDS-PAGE. Proteins were transferred to a PVDF membrane (Millipore), blocked by blocking buffer (Rockland) and incubated with primary and IRdye-labeled secondary antibodies (1:10K1-25K dilution; Licor). Blots were visualized on an Odyssey scanner (Licor). Anti-p24 antibody used for HIV-1 capsid detection was AG3.0 (NIH-ARP-4121)[43] and 183-H12-5C (NIH-ARP-3537), both used at 1:1000 dilution.

Labeling and immunoprecipitation for mass spectrometry was performed using the SILAC Protein Quantitation-Trypsin kit and MS-compatible Magnetic IP kit from Pierce (Thermo) according to manufacturer's instructions. Cells were cultured in light medium with 10% dialyzed FBS and transduced with viruses also produced in light medium. The virus used was VSV-G-pseudotyped, NL4.3-Firefly reporter virus carrying 10 amino acids from the p6 region of SIVmac (pNL-luc3-SIVp6[17–26])[44]. Infection was performed in the presence or absence of NVP (25 μM) or CHX (100 μg/ml) at an MOI of 0.5 by spinoculation. The conditions were: Mock (no virus), virus (no drug), virus + NVP, and virus + NVP + CHX. At the time of transduction, cells were switched to heavy medium. After incubation for 18 h, cells were washed extensively with PBS or PBS + CHX (50 μg/ml) to remove all heavy media and lysed in IP-MS cell lysis buffer with protease inhibitors on ice. Immunoprecipitation was performed with an anti-p24 antibody (183-H12-5C; NIH-ARP-3537) using Pierce Protein A/G magnetic beads. Eluted proteins were analyzed by LC-MS. Samples were prepared and analyzed in biological triplicates.

## Liquid chromatography and mass spectrometry

Peptides were analyzed on an Evosep One liquid chromatography system coupled online via the CaptiveSpray source to a timsTOF HT mass spectrometer (Bruker Daltonics). Peptides were manually loaded onto Evotips Pure (Evosep) and separated using the 30 samples per day (SPD) method on the respective performance column (15 cm × 75 μm, 1.9 μm, Evosep). Column temperature was kept at 40 °C using a column toaster (Bruker Daltonics) and peptides were ionized using electrospray with a CaptiveSpray emitter (10 μm i.d., Bruker Daltonics) at a capillary voltage of 1400 V. The timsTOF HT was operated in ddaPASEF mode in the m/z range of 100–1700 and in the ion mobility (IM) range of 0.65–1.35 Vs cm$^{-2}$[45]. Singly-charged precursors were filtered out based on their m/z-ion mobility position. Precursor signals above 2500 arbitrary units were selected for fragmentation using a target value of 20,000 arbitrary units and an isolation window width of 2 Th below 700 Da and 3 Th above 700 Da. Afterwards, fragmented precursors were dynamically excluded for 0.4 min. The collision energy was decreased as a function of the IM from 59 eV at 1/K0 = 1.6 Vs cm$^{-1}$ to 20 eV at 1/K0 = 0.6 Vs cm$^{-1}$. One cycle consisted of 10 PASEF ramps.

## MS data analysis

The LC-IMS-MS/MS data were analyzed using FragPipe (version 20.0)[46]. Spectra were searched using MSFragger against the protein sequences of the human proteome (UP000005640, UniProtKB) and of HIV-1 (NL4-3 e- r- Fluc [ARP-3418] with a modified SIV p6 between aa 17–26) with a precursor and fragment mass tolerance of 20 ppm, strict trypsin specificity (Lysine: K, Arginine: R) and allowing up to two missed cleavage sites. Cysteine carbamidomethylation was set as a fixed modification and methionine oxidation, N-terminal acetylation of proteins as well as heavy labeling of lysine and arginine (K + 8.014199 Da, R + 10.008269 Da) as variable modifications. Search results were validated using Percolator with MSBooster enabled rescoring and converged to false discovery rates of 1% on all levels.

Proteins were quantified using IonQuant based on peptides consistently identified in all replicates and requiring at least 2 peptides per protein. The mass spectrometry proteomics data have been deposited to the ProteomeXchange Consortium via the PRIDE partner repository with the dataset identifier PXD046777.

## p24 ELISA

p24-CA concentrations of viral stocks were determined by a home-made ELISA[43]. Briefly, 96-well plates were coated overnight at 4 °C with the AG3.0 anti-p24 antibody diluted 1:100 in carbonate/bicarbonate buffer (Sigma). The next day, plates were washed three times with wash buffer (PBS + 0.05% Tween), blocked with PBS + 2% milk powder at 37 °C for 1 h and washed again three times. Meanwhile, the viral supernatants were inactivated by incubating with a final concentration of 0.2% Tween for 10 min at room temperature. Serial dilutions of viral supernatants in dilution buffer (PBS + 2% milk powder + 0.05% Tween) were pipetted into the wells and incubated at 37 °C for 1 h. After three washes, the primary antibody in the form of pooled HIV+ serum diluted 1:10,000 was added to the wells and incubated at 37 °C for 1 h. Following another three washes, the secondary antibody anti-human IgG-HRP (Sigma) was added at 1:1000 dilution. After three more washes, the substrate solution (12.5 ml phosphate/citrate buffer + 1 OPD tablet (5 mg; Sigma) + 12 µl 30% $H_2O_2$ solution) was added and incubated at room temperature for 10 min. Reactions were stopped by adding 5% sulfuric acid ($H_2SO_4$). Absorbance was measured at 492 and 620 nm.

## Statistics

All statistics analyses were performed using GraphPad Prism 9. For each figure, the numbers of biological replicates are as follows: Fig. 1B: $n = 7$, Fig. 1C: $n = 6$, Fig. 1D: $n = 3$, Fig. 1E, F: $n = 2$, Fig. 2B: $n = 3$, Fig. 2C: $n = 7$, Fig. 2D, E: $n = 6$, Fig. 2F: $n = 3$, Fig. 2G: $n = 6$, Fig. 3A: $n = 3$–$6$, Fig. 3B: $n = 1$, Fig. 3C, D: $n = 6$, Fig. 3E: $n = 5$, Fig. 3F: $n = 6$–$12$, Fig. 3G: $n = 3$–$6$, Fig. 3H: $n = 6$, Fig. 3I, J: $n = 3$, Fig. 4B: $n = 5$, Fig. 4C: $n = 2$, Fig. 4D: $n = 3$, Fig. 4E: $n = 3$, Fig. 4F: $n = 4$, Fig. 5A: $n = 5$–$12$, Fig. 5B: $n = 10$, Fig. 5C: $n = 3$, Fig. 5D: $n = 2$, Fig. 5E: $n = 6$, Fig. 5F: $n = 12$. Supplementary Fig. 1A: $n = 2$–$4$, Supplementary Fig. 1B, C: $n = 3$, Supplementary Fig. 2A: $n = 2$, Supplementary Fig. 3A, B: $n = 3$.

## Reporting summary

Further information on research design is available in the Nature Portfolio Reporting Summary linked to this article.

## Data availability

The mass spectrometry data generated in this study have been deposited in the public database PRIDE (https://www.ebi.ac.uk/pride/) under the accession number PXD046777. Source Data are provided with this paper.

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

## Acknowledgements
We thank members of the Cingöz Lab and our colleagues in FG18 for helpful discussions. We also thank Christine Goffinet and her group for valuable input and materials, as well as Igor Minia for help with polysome fractionation. We thank Kornelia Gericke and Sabina Reichert for excellent laboratory management. This work was funded by priority program on "Innate sensing and restriction of retroviruses" (SPP1923) of the German Research Council (DFG; CI305-1/2, OC and BA1897/3-2, NB), intramural funds from the Robert Koch Institute (FG18), as well as the KT Boost Fund (KT36, OC) by the Klaus Tschira Foundation and the German Scholars Organization (GSO).

## Author contributions
J.K., L.K., M.S., N.A., J.D. and O.C. performed experiments, analyzed and visualized data. N.B. provided resources. O.C. wrote the manuscript. N.A. and N.B. reviewed the manuscript. O.C. acquired funding, supervised and managed the project.

## Funding

## Competing interests
The authors declare no competing interests.
