## [Peer Review File · Nature Communications]

Reviewers' Comments:

Reviewer #1:

Remarks to the Author:

None

Reviewer #2:

Remarks to the Author:

This is a very interesting and provocative paper. Evidently, and as presented, the work can be viewed as somewhat anti-dogmatic – which naturally places a high burden of proof upon the authors. The work is notably assiduous and well-controlled throughout, making good use of a variety of vectors/viruses and genetically altered versions thereof. Accordingly, aggregating the complimentary lines of evidence leads one to agree with the key conclusion of the study – that retrovirus genomes can be translated in newly infected (challenged) cells irrespective of the productive/ non-productive outcome of the infection. As such, the work convincingly challenges accepted retroviral dogma, and will be of general interest to virologists.

Specific comments and suggestions.

1. All vector/virus challenges appear to employ a single dose. The authors should provide an indication of the inocula used (e.g., ng of p24 per million target cells). Also note point 6 below.
2. Comparing 2D with 2C, can the authors comment on why the levels of Nluc increase more obviously with time in D than C? E.g. illustrated by no drug condition in the first 4 hours.
3. Panels 2D and 3B. The signals increase over time in the presence of CHX, what is the explanation? Is this reflecting continuous virus uptake and entry? Would washing the challenged cultures after inoculation minimise this?
4. Provide a description for how the experiments with NL43DD-Nluc were executed. Presumably a helper provirus was used to generate input virus?
5. Gag proteins detected in 3C and 3D. In the former, the more prominent Gag species is p55, in the latter it is p24. What is the explanation?
6. One can imagine that readers will want to know about the levels of input inocula used for these studies (refer to point 1). The study would be strengthened by providing this information. For instance, through per cell quantification of input viral RNA, reverse transcripts and proviruses for one set of challenges – e.g., for the infections shown in 3B with the possible addition of a +NVP virus challenge.
7. As a further experimental verification of viral protein production in the absence of cDNA synthesis or transcription, have the authors considered other methodologies for measuring nascent Gag translation, such as S-35 pulse-chase or a SILAC-mass spectrometry based approach?

Reviewer #3:

Remarks to the Author:

Here, Koppke et al. use destabilized nano-luciferase (DD-Nluc) to study the capacity of HIV-1 genomes and HIV-derived vectors to be translated in target cells as incoming mRNAs (i.e., independently of reverse transcription). The DD-Nluc strategy is elegant, and reduces potential for non-specific signal

from packaged reporter protein that may have confounded prior analyses. Figure 2 data is convincing that translation of a packaged transgene/genome can occur in target cells, and suggests that initiation is positionally dependent (i.e., I interpret the differences to 2C/D vs. 1E as meaning that the start codon would need to be proximal to the 5'UTR). A DD-N-luc HIV-1 reporter virus (Figure 3) is engineered and used to show that incoming direct translation can also occur in the context of a full-length HIV-1 genome, a key finding, with viral RNA confirmed as associated with polysomes. Finally, Figure 4 presents data suggesting a correlation between capsid stability and the potential for genome translation. Together, the authors provide data in support of a model wherein it may be possible for HIV-1 genomes to be translated during early infection to yield viral gene products (Gag proteins), although the significance of direct translation is not yet addressed. In general this is a well-designed study with interesting results. However, I do have concerns regarding the model and some of the controls, outlined below.

Major comments:

1. The paper is pitched as countering dogma but, as acknowledged by the authors, (1) direct translation of retroviral packaged mRNA is not a novel concept, especially in the context of gene therapy (e.g., see cited references 25, 26), so that some potential for viral gene expression isn't that surprising; and (2) for natural infection, genome translation would most likely occur at the expense of reverse transcription, so that translation would be predicted to represent a dead end pathway. Together, while the DD-Nluc data strengthen the notion that direct translation can occur, its significance remains unknown and, at present, it seems like odds are that this is an unlikely event during replication. I do like the idea that aberrant early Gag synthesis tied to the timing of uncoating could trigger immune signaling relevant to pathogenesis, but this is speculative. [SEP]
2. A general question is how much virus is used in these experiments. The particle to PFU ratio for HIV-1 is thought to be relatively low in general. It is interesting to consider that direct translation could be one of those reasons. Surprisingly, MOI is not mentioned at all in this paper. The dogma-relevant conclusions would be stronger if there was a way to confirm the conditions approximate the physiological.
3. Some of the data in Figure 3 seem unexpected, specifically panels 3C and 3D that show detection of Gag p24 capsid subunits. p24 should only be produced if (1) both Gag and Gag-Pol are being synthesized (Gag-Pol being the source of viral protease) and (2) virions are being produced (late assembly is when protease is activated). Sufficient Gag and Gag-Pol synthesis to achieve virion production would seem unlikely under these conditions (especially in the presence of NVP). Based on this issue the experiments tracking Gag synthesis may have been misinterpreted, evidence for Gag translation needs to be strengthened.
4. Figure 4 would benefit from controls demonstrating that the P38A genomes are equivalently packaged and truly more often translated per virion relative to WT. I interpreted the 4F immunoblot as indicating that equivalent levels of p24 were added, but it is not described, and even if that is the case more should be done here to strengthen the link to capsid integrity. I do like the idea.
6. I had general concerns regarding statistical comparisons throughout, there is not much description and error bars tend to be tight- have sufficient biological replicates been performed to allow for robust conclusions?

Minor comments

1. Figure 1- the WPRE is featured in these constructs but not in the Figure 2 vectors- should be explained. Related, some of the nomenclature (e.g., why some vectors are called "WPRES" and others are not even though they carry a WPRES) is confusing.

2. Figure 2. 2G lacks a control to show that ActD is working.

3. Figure 4D- the authors suggest that Δ psi is not affected by CHX. That does not look accurate, drop seems similar to WT.

It was very encouraging to read the overall positive comments on our manuscript. We thank the reviewers for sharing their time and expertise. We have performed many additional experiments as recommended and revised our manuscript, which we believe has significantly improved its quality. All of the comments are addressed point by point below.

REVIEWER COMMENTS

Reviewer #2 (Remarks to the Author):

This is a very interesting and provocative paper. Evidently, and as presented, the work can be viewed as somewhat anti-dogmatic – which naturally places a high burden of proof upon the authors. The work is notably assiduous and well-controlled throughout, making good use of a variety of vectors/viruses and genetically altered versions thereof. Accordingly, aggregating the complementary lines of evidence leads one to agree with the key conclusion of the study – that retrovirus genomes can be translated in newly infected (challenged) cells irrespective of the productive/ non-productive outcome of the infection. As such, the work convincingly challenges accepted retroviral dogma, and will be of general interest to virologists.

Specific comments and suggestions.

1. All vector/virus challenges appear to employ a single dose. The authors should provide an indication of the inocula used (e.g., ng of p24 per million target cells). Also note point 6 below.

Infection with serial dilutions of our reporter virus showed that a CHX-sensitive signal can be detected using as low as 0,0061 ng (~6 pg) p24 per 100K cells (Fig. 2C). Notably, in case of infection with input virus doses above 4.5 ng or below 0,165 ng p24 per 100K cells, the difference between the background signal due to protein transfer and the test signal due to de novo translation actually decreased (52-fold vs. 12-fold, see figure on the right). We have also added text to indicate the inocula used, wherever relevant. Based on the results of serial dilutions, all experiments were then performed with 1-4 ng p24 per 100K cells (lines 101-104).

2. Comparing 2D with 2C, can the authors comment on why the levels of Nluc increase more obviously with time in D than C? E.g. illustrated by no drug condition in the first 4 hours.

This phenomenon can be explained by the difference in background levels between DD-containing and -lacking reporters. Both the amount of virion-packaged reporter, as well as the half-life of those that are transferred to the recipient cells are greatly reduced by the presence of a DD domain (Fig. 1D-E), such that the de novo translation from the incoming viral RNA is masked less by the background signal, resulting in a sharper increase. The removal of the initial inoculum also does not change this phenotype (Fig 2D-E vs. Fig 2F-G).

3. Panels 2D and 3B. The signals increase over time in the presence of CHX, what is the explanation? Is this reflecting continuous virus uptake and entry? Would washing the challenged cultures after inoculation minimise this?

Due to the short transduction times used in our time-course assays (1-8 hours), we initially opted to keep the virus on the cells for the duration of the experiment. As suggested by the reviewer, the increased signal in CHX reflects continued virus uptake by the cells. We did the experiment recommended by the reviewer, where cells were spinoculated for one hour and the virus was removed afterwards. Under these conditions, there is no increase in the CHX samples. These data have been added in Fig 2F-G.

4. Provide a description for how the experiments with NL43DD-Nluc were executed. Presumably a helper provirus was used to generate input virus?

That is correct. NL43-DD-Nluc cannot produce Gag/Pol due to the DD-Nluc insertion, so viruses were produced using a packaging plasmid. We added a description to both the results and methods sections to reflect this point (lines 155-156 and 312-313).

5. Gag proteins detected in 3C and 3D. In the former, the more prominent Gag species is p55, in the latter it is p24. What is the explanation?

We were also surprised about the difference between the p55 vs p24 in these experiments. The experiment in former 3C is immunoprecipitation of Gag using a rabbit polyclonal anti-p17+24+55 [sic] antibody (Abcam, ab63917) in cells infected with NL43-Firefly virus in the presence of NVP, whereas the experiment in former 3D is a direct western blot (not IP) of cells infected with IIB catalytic RT mutant virus (YMDD>YAAA). The polyclonal antibody used in the former experiment is able to pull down Gag due to its binding to MA as well as CA, which is not the case in a direct western blot. The two blots are therefore not directly comparable, as they reflect different experimental setups. However, also based on point 3 raised by Reviewer #3, we removed this figure as it creates some confusion and we are not entirely sure how this processing takes place. As the reviewer correctly states, the production of p24 would require enough levels of GagPol with protease activity to process Gag, and we do not have any evidence to suggest that this occurs. As further proof that viral Gag is produced in the absence of RT, we instead performed metabolic labeling (SILAC) and immunoprecipitation followed by mass spectrometry, and detected peptides mapping to different regions of Gag (also see point 7 below). These results are now shown in Fig. 4D.

6. One can imagine that readers will want to know about the levels of input inocula used for these studies (refer to point 1). The study would be strengthened by providing this information. For instance, through per cell quantification of input viral RNA, reverse transcripts and proviruses for one set of challenges – e.g., for the infections shown in 3B with the possible addition of a +NVP virus challenge.

Thank you for the suggestion. As also mentioned in our answer to point 1, experiments with serial dilutions of reporter viruses showed a CHX-sensitive signal with as little as 6 pg p24 per 100K cells, which corresponds to ~10K RNA copies/100K cells (0,1 RNA copies/cell). At the typical amounts used for most of our experiments (1-4 ng p24), it corresponds to 1,9-7,8 RNA copies/cell. As our Nluc reporter viruses do not have express viral structural proteins or a fluorescent reporter, it is not possible to calculate infectious units or MOI for them. For infections with (near) full-length viruses used in immunoprecipitation and SILAC-MS, we did perform this calculation and added the information in the text (MOI 0.5-1, depending on the experiment; lines 163, 168, 178, 381). We also quantified early/late RT products and 2-LTR circles. It should be noted that for early & late RT products, we often get a high background signal. As infection with 4 ng p24 per 100K cells failed to

yield a signal above the background, we performed the quantification of using 10x more virus than normally used (40 ng p24), which corresponds to an MOI = 1. These data have been added in Supp. Fig. 2A. Unfortunately, our efforts to quantify integrated proviruses during infection of 293T cells with our reporter viruses have not been successful, even using 10x more virus and after 24-48 hours post-infection (data not shown).

7. As a further experimental verification of viral protein production in the absence of cDNA synthesis or transcription, have the authors considered other methodologies for measuring nascent Gag translation, such as S-35 pulse-chase or a SILAC-mass spectrometry-based approach?

Both great ideas, thanks. As our institute no longer carries a radiation license, we opted for the SILAC-MS experiment. Briefly, cells in light medium were challenged with virus also produced in light medium. At the time of transduction, the culture was switched to heavy medium containing drugs; such that all newly-produced proteins would be heavy and all existing proteins light. We then lysed the cells, performed a pull-down with an anti-p24 antibody and analyzed the contents of the pull-down by mass spec. We detected heavy peptides corresponding to Gag in the virus-infected samples in the presence of NVP, which went away with CHX treatment. We included these data in Fig. 4D. The raw data will be uploaded on the public server PRIDE (Proteomics Identifications Database) hosted by EMBL-EBI.

Reviewer #3 (Remarks to the Author):

Here, Koppke et al. use destabilized nano-luciferase (DD-Nluc) to study the capacity of HIV-1 genomes and HIV-derived vectors to be translated in target cells as incoming mRNAs (i.e., independently of reverse transcription). The DD-Nluc strategy is elegant, and reduces potential for non-specific signal from packaged reporter protein that may have confounded prior analyses. Figure 2 data is convincing that translation of a packaged transgene/genome can occur in target cells, and suggests that initiation is positionally dependent (i.e., I interpret the differences to 2C/D vs. 1E as meaning that the start codon would need to be proximal to the 5'UTR). A DD-N-luc HIV-1 reporter virus (Figure 3) is engineered and used to show that incoming direct translation can also occur in the context of a full-length HIV-1 genome, a key finding, with viral RNA confirmed as associated with polysomes. Finally, Figure 4 presents data suggesting a correlation between capsid stability and the potential for genome translation. Together, the authors provide data in support of a model wherein it may be possible for HIV-1 genomes to be translated during early infection to yield viral gene products (Gag proteins), although the significance of direct translation is not yet addressed. In general this is a well-designed study with interesting results. However, I do have concerns regarding the model and some of the controls, outlined below.

As many of the points raised by the reviewers are similar, we sometimes refer to our answers to Reviewer #2 above to minimize repetition.

Major comments:

1. The paper is pitched as countering dogma but, as acknowledged by the authors, (1) direct translation of retroviral packaged mRNA is not a novel concept, especially in the context of gene therapy (e.g., see cited references 25, 26), so that some potential for viral gene expression isn't that surprising; and (2) for natural infection, genome translation would most likely occur at the expense of reverse transcription, so that translation would be predicted to represent a dead end pathway. Together, while the DD-Nluc data strengthen the notion that direct translation can occur, its significance remains unknown and, at present, it seems like odds are that this is an unlikely event during replication. I do like the idea that aberrant early Gag synthesis tied to the timing of uncoating could trigger immune signaling relevant to pathogenesis, but this is speculative.

Ref. 25 (Galla et al. 2004) is a study on MLV-based gene-therapy vectors, which differs from the work presented here in several aspects: 1) MLV vs. HIV-1 in this study, 2) For MLV, RT is completed and the capsid dissociates in the cytoplasm that would allow easier access to ribosomes as opposed to HIV-1 capsids, which were shown to dissociate at the nuclear pore or inside the nucleus, 3) Though the vectors used in the study did have PBS mutations, RT activity was not blocked by RT inhibitors or catalytic site mutations, 4) Passive delivery of virion-packaged protein or DNA was not ruled out (see refs. 1-6), 5) The only genomes used were modified minimal vectors, and 6) no actual viral protein synthesis was shown. Ref 26 (Counsell et al. 2021) is a study on lentivirus-based gene-therapy vectors, with substantial genomic structural rearrangements and insertions so as to specifically drive translation from a highly-modified packaged RNA, for instance by inserting an IRES or a WPRE element upstream or downstream of an EGFP reporter gene, respectively, and moving the R-U5 regions further towards the 3' end of the genome.

In spite of such previous work in the field of gene therapy, we believe the concept of direct expression from incoming retroviral genomes is an aspect of retrovirus literature that has not been fully recognized. To our knowledge, direct expression from incoming retroviral RNA has not been shown for the production of authentic viral proteins to date, or with all of the appropriate controls presented in our paper, or with an unmodified / minimally modified (near-full-length) HIV-1 genome that is not in the form of a completely remodeled lentiviral vector. While the use of such vectors as delivery vehicles has been previously considered in the literature, the occurrence of this process in the context of an infection, regardless of whether reverse transcription can take place, and a discussion on the potential consequences of this process is lacking.

2. A general question is how much virus is used in these experiments. The particle to PFU ratio for HIV-1 is thought to be relatively low in general. It is interesting to consider that direct translation could be one of those reasons. Surprisingly, MOI is not mentioned at all in this paper. The dogma-relevant conclusions would be stronger if there was a way to confirm the conditions approximate the physiological.

Thank you for the suggestion and please see our answers to points 1 and 6 from Reviewer #2, regarding p24/RNA/MOI calculations. These data have been added. Whenever reporter viruses were used that do not encode viral structural proteins or a fluorescent gene, it was not possible to calculate infectious units/MOI, so in these instances we quantified the p24 and RNA amounts. Whenever we used (near) full length viruses, we stated the MOI; for IP and SILAC/MS experiments the MOI were 1 and 0.5, respectively. We also mention the high particle to PFU ratio in the discussion section (lines 265-268). It is not entirely clear what kind of an inoculum is encountered in a physiological setting; during an initial infection it's expected to be low, whereas during replication in a lymph node involving cell-to-cell transmission it could be rather high (e.g. Duncan et al. JVI 2014, Del Portillo et al. JVI 2011, or the review Agosto et al. Trends Micro 2015).

3. Some of the data in Figure 3 seem unexpected, specifically panels 3C and 3D that show detection of Gag p24 capsid subunits. p24 should only be produced if (1) both Gag and Gag-Pol are being synthesized (Gag-Pol being the source of viral protease) and (2) virions are being produced (late assembly is when protease is activated). Sufficient Gag and Gag-Pol synthesis to achieve virion production would seem unlikely under these conditions (especially in the presence of NVP). Based on this issue the experiments tracking Gag synthesis may have been misinterpreted, evidence for Gag translation needs to be strengthened.

We agree; please also see our answer to points 5 and 7 of Reviewer #2 above. We removed this figure and performed SILAC-MS instead to provide further evidence for de novo Gag production.

4. Figure 4 would benefit from controls demonstrating that the P38A genomes are equivalently packaged and truly more often translated per virion relative to WT. I interpreted the 4F immunoblot as indicating that equivalent levels of p24 were added, but it is not described, and even if that is the case more should be done here to strengthen the link to capsid integrity. I do like the idea.

Thank you for the suggestion. we have now added data wherein viruses with different capsids (WT, P38A and E45A) were normalized in several different ways: by p24 content, by RNA copies, by volume, and by virion-packaged Nluc protein (Supp. Fig. 3) As there is some variation based on each assay (see ref. 20), we believe the normalization method most informative for our purposes is the packaged Nluc protein, such that all CHX-treated samples yield equivalent results. We include the collective data in Fig. 5F, which confirms that the incoming genome found in a hypostable capsid (P38A) is translated more efficiently overall. The hyperstable capsid (E45A) differs slightly based on the normalization method is used; however, in the collective data, the translation efficiency is below that of WT capsid, which also fits well with the findings described in this paper.

6. I had general concerns regarding statistical comparisons throughout, there is not much description and error bars tend to be tight- have sufficient biological replicates been performed to allow for robust conclusions?

In each figure, we have included biological replicates as individual data points, explained what is being shown in the graphs (e.g. mean/median, SD/SEM), described the statistical tests performed and named the correction methods used.

Minor comments

1. Figure 1- the WPRE is featured in these constructs but not in the Figure 2 vectors- should be explained. Related, some of the nomenclature (e.g., why some vectors are called "WPRE" and others are not even though they carry a WPRE) is confusing.

Apologies for the confusion. All constructs are based on the pLVX-IRES-Neo vector (Clontech). Initially, we tried to keep the names and drawings as simple as possible (didn't list every element in the plasmid), but we realize this may have been confusing. The first two constructs, previously named CMV-(DD)-Nluc are based on the complete original vector, where the reporter gene is cloned into the MCS (now constructs I and II). The last two, previously named (DD)Nluc-WPRE, have the CMV promoter, IRES and Neo removed from them (now constructs III and IV). To avoid any confusion, we prepared new drawings that show the other elements and changed the names of the constructs to pLVX-(DD)-Nluc and pLVX-(DD)-Nluc ΔCIN (for deletions in CMV, IRES and Neo). As these names are rather long, we refer to them in their respective roman numerals. These constructs cannot support direct expression from the incoming viral genome, as they contain multiple start/stop codons in the sequence upstream of the reporter, as well as splice donor/acceptor sites. The vectors in Fig. 1 were used to establish and optimize the ProteoTuner system. A detailed explanation of the cloning strategy can be found in the Methods section (lines 295-304).

The vectors in Fig. 2 and those used in the rest of the manuscript do not contain a WPRE. To assay translation from the incoming genomes, we removed any extra sequences that may interfere with translation. All experiments to support the notion of direct translation of incoming retroviral genomes were performed without a CMV promoter or an IRES element or a Neo resistance gene or a WPRE element. We realize this may not have been clearly explained in the paper, so we added text to Results, Methods and figure legends clarify this point (Lines 95-96, 301-302 and 460-462).

2. Figure 2. 2G lacks a control to show that ActD is working.

Fair point; we have included data to show ActD is working. Transfection of 293T cells with a CMV-driven Nluc vector in the presence of either ActD (2 $\mu\text{g}/\text{ml}$) or CHX (100 $\mu\text{g}/\text{ml}$) blocks expression from the plasmid at 4 hours. Yet, during infection with our reporter viruses under the same drug concentrations, translation from the incoming retroviral genome remains unaffected by ActD; if anything, it is even up by 3-fold, while CHX reduces it to $\sim 7\%$ of untreated levels. These results are now included in Fig. 3F-G.

3. Figure 4D- the authors suggest that Δpsi is not affected by CHX. That does not look accurate, drop seems similar to WT.

The difference between CHX and no CHX was actually quite different between WT (~ 22 -fold) and Δpsi (2.8-fold; the graph is log scale). The psi deletion we introduced initially [741-809 nt] largely impaired expression from the plasmid; the result being less Nluc protein packaged into virions to start with compared to WT, as seen in the difference between CHX treated samples (former Fig. 4D, grey bars). Thus, we believe it is not a fair comparison to check the impact of this particular deletion on the incoming viral RNA translation, since the expression potential itself is altered, regardless of whether the RNA can be packaged into particles or not. To avoid confusion, we removed this figure and instead introduced a smaller psi mutation [750-787 nt] that does not prevent expression and even increases it by 2.5-fold (Fig. 5B). RT-qPCR confirmed less viral RNA packaging into particles (Fig. 5A), and infection resulted in less newly-produced reporter signal, supporting our data (Fig. 5C). Please also see the data on the right side, where cells were transfected with the two psi deletion constructs to quantify expression.

OTHER CHANGES INTRODUCED

The nature of the catalytic RT mutations in different viral constructs was expanded. To demonstrate the lack of RT activity in our RT mutants, we added PERT assay data (Supp. Fig. 1A-B). To ensure our results were not due to incomplete inhibition with NVP at the concentrations used, experiments with three different RT inhibitors (NVP, EFV, TAF) were performed at a range of concentrations (1-100 μM), the majority of which did not alter the phenotype. Only EFV at 100 μM killed all signal, thus additional data was added to demonstrate that the drug is toxic to cells at this concentration and inhibits the expression of any transfected plasmid, not just the incoming viral genomic RNA translation. These data are now in Fig. 3H and Supp Fig. 1C-D.

The main text was divided into the following sections: Introduction, Results, Discussion, Methods. Several supplementary figures were added. All changes are marked red in the revised manuscript.

Reviewers' Comments:

Reviewer #2:

Remarks to the Author:

Köppke and colleagues have responded robustly to all of my initial comments, and the manuscript has been improved yet further. The validation of new protein synthesis in the presence of NVP using a SILAC-based approach serves as additional convincing evidence that incoming HIV genomes can be translated. This work is of high quality and has been scrupulously well-controlled – I believe it will be of broad interests to virologists and should now be published in Nature Communications.

Reviewer #3:

Remarks to the Author:

The authors have addressed my concerns.